# Factors Predicting Sexual Satisfaction of Thai Women with Cervical Cancer After Complete Treatment at Least One Year and Their Spouses

**DOI:** 10.3390/healthcare13020112

**Published:** 2025-01-09

**Authors:** Wunwisa Bualoy, Bualuang Sumdaengrit, Tiraporn Junda, Arb-aroon Lertkhachonsuk

**Affiliations:** Faculty of Medicine Ramathibodi hospital, Mahidol University, Nakhon Pathom 10400, Thailand; wunwisa.bul@mahidol.ac.th (W.B.); tiraporn.jun@mahidol.ac.th (T.J.); arbaroon@hotmail.com (A.-a.L.)

**Keywords:** sexual satisfaction, cervical cancer, post treatment, spouses, Thailand

## Abstract

Background: Sexual satisfaction is vital for the sexual health and well-being of both cervical cancer patients and their spouses. Sexual dissatisfaction can arise from negative treatment, making it important to examine the factors that influence sexual satisfaction. Objective: The purpose of this study was to explore the factors affecting the sexual satisfaction of Thai women with cervical cancer after complete treatment and their spouses. Materials and Methods: A predictive correlational study was conducted with 100 couples using convenience sampling. The study was based on the ecological theory framework. Data were collected from April 2023 to January 2024 in two tertiary hospitals through five questionnaires: a demographic questionnaire, the Thai Hospital Anxiety and Depression Scale, the Charlson Comorbidity Index, the Couple Relationship Scale, Natsal 2000, and the New Sexual Satisfaction Scale. Data analysis was performed using multiple linear regression. Results: The multiple regression analysis showed that sexual difficulties and duration of marriage predicted 25.6% of the variance in sexual satisfaction among women with cervical cancer. The duration of marriage (b = −0.48, *p* = 0.012) had a stronger impact than sexual difficulties (b = −2.82, *p* = 0.005). Sexual difficulties and couple relationships accounted for 34.9% of the variance in sexual satisfaction for spouses. Sexual difficulties (b = −3.13, *p* = 0.004) had a greater effect than couple relationships (b = 0.30, *p* = 0.003). Conclusions: Healthcare providers should promote constructive communication between couples through open, respectful, and supportive dialogue to strengthen their relationship, improve mutual understanding, address sexual difficulties, and enhance sexual satisfaction for women with cervical cancer after treatment and their spouses. The findings results can inform future intervention research aimed at improving sexual satisfaction in this population one-year post treatment.

## 1. Introduction

Cervical cancer is a significant gynecological condition that can be prevented and cured if detected early [1]. With effective screening and various treatment options like chemotherapy, radiotherapy, and surgery, the 5-year survival rate at early stages can reach 91% [2]. In 2022, a total of 1,473,427 women were diagnosed with gynecological cancer, with 44.95% having cervical cancer, while 680,732 women died from gynecological cancer, and 51.28% of these deaths were attributed to cervical cancer. The Association of Southeast Asian Nations (ASEAN), consisting of 10 countries including Thailand, reported 69,000 new cervical cancer cases and 38,000 deaths in 2020, representing 11.3% of the global incidence and 11.3% of mortality from cervical cancer worldwide [3,4,5,6]. Cervical cancer cases and deaths in ASEAN are projected to rise, with an estimated 99,000 new cases and 63,000 deaths expected by 2024 if effective prevention measures are not introduced promptly [3,4,5,6]. In Thailand, cervical cancer is the second most common cancer after breast cancer, accounting for approximately 13.8% of new cases in 2021 [7]. The 10-year survival rate is about 88% for stage I, whereas it drops below 50% for advanced stages [7]. Despite higher survival rates, different types of cervical cancer treatment can cause different side effects, including issues with sexual satisfaction. Sexual satisfaction is defined as how someone feels about their sexual relationship, based on their own thoughts and feelings about the positive and negative aspects of it, such as feelings of pleasure, desire, and emotional closeness with their partner [8].

Generally, cervical cancer survivors often experience a significant decline in sexual function [9]. They may face issues like reduced libido, difficulty with arousal and orgasm, increased vaginal dryness and stenosis, decreased lubrication and sensation, and pain during sexual intercourse [9]. In addition, cervical cancer survivors are typically younger, with an average diagnosis age of 49; thus, their intimate relationships can impact both themselves and their spouses [10]. Previous studies revealed that both survivors and their partners experienced sexual distress; specifically, survivors reported sexual dissatisfaction due to vaginal sexual symptoms and pain before, during, and after sexual intercourse [11]. Moreover, they noticed negative effects on their sexual functioning, sex lives, and their relationships with partners [12]. In Thailand, research on cervical cancer survivors after treatment is limited. A 2022 study found that nearly 90% of Thai women experienced sexual dysfunction after treatment [13]. However, this study focused on gynecologic cancer in general on sexual dysfunction and did not include partners or spouses, as in the present study [13]. In addition, Thai cultural norms that restrict women’s sexual expression and discourage open discussion of sexual desires and satisfaction, even with spouses, pose challenges, particularly for cervical cancer patients, in understanding how limited sexual expression affects them after cervical cancer treatments [14,15]. Therefore, it is important to investigate sexual satisfaction among patients and their spouses, along with the dynamics of their relationship. This investigation can be framed within the ecological theory, which provide a comprehensive understanding of sexual satisfaction by examining how individual, relational, and societal factors are interconnected and influence sexual satisfaction.

The ecological theory, developed by Henderson and colleagues in 2009, includes four levels: microsystem, mesosystem, exosystem, and macrosystem [16,17]. In this study, we investigate the factors that contribute to sexual satisfaction between women with cervical cancer and their spouses, so the microsystem and mesosystem were chosen as they are directly linked to sexual satisfaction in couples [18,19,20,21,22,23]. The microsystem encompasses individual aspects like age, anxiety, depression, and comorbidities, while the mesosystem addresses relationship factors such as sexual couple dynamics, sexual difficulties, and marriage duration [19,20,22,23]. In fact, all of these factors interact to influence both couples’ satisfaction and the overall quality of their relationship, particularly in the context of sexual satisfaction [24]. Please see Figure 1 for details. On the other hand, the exosystem, which includes factors like socioeconomic status and social support, does not have a direct impact on sexual satisfaction. Similarly, the macrosystem, which involves broader societal and institutional factors, has little evidence connecting it to sexual satisfaction. Both levels are considered the distal factors, therefore, they were excluded from this study.

The purpose of this study was to investigate the factors—such as age, anxiety, depression, comorbidity, couple relationship, sexual difficulties, and duration of marriage—that predict the sexual satisfaction in Thai women with cervical cancer after completing treatment, as well as their spouses.

## 2. Materials and Methods

### 2.1. Study Design

A descriptive, cross-sectional study was conducted to quickly explore the issue and gain insights into factors that may be linked to sexual satisfaction. Data were collected using a self-administered, paper-based questionnaire. The questionnaire was provided at the gynecology outpatient clinic, where participants had appointments with their doctors. A private room was provided for the participants to complete the questionnaire either before or after their appointment. Before completing the questionnaire, participants had to agree to the research’s purpose, the voluntary nature of their involvement, and the assurance of data privacy.

To minimize potential self-report biases, the principal investigator (PI) provided cultural sensitivity training for nurses involved in recruitment, ensuring respectful interaction, and an understanding of Thai cultural norms. During data collection, the PI used non-judgmental language, and participants were assured that their data would remain confidential and anonymous.

### 2.2. Samples/Participants

The target population of this study was Thai women who had undergone treatment for cervical cancer for at least one year and their spouses, recruited from two tertiary hospitals in Bangkok, Thailand. Data were collected from April 2023 to January 2024. Participants had to meet the following inclusion criteria: (1) first diagnosed with cervical cancer at stage I, II, or III; (2) aged 18–59 years; (3) have received surgery, radiotherapy, and/or chemoradiotherapy; (4) completed treatment between one and five years ago; (5) be able to read and write the Thai language; (6) have a spouse or partner who lived with them prior to the diagnosis; and (7) have engaged in sexual intercourse (SI) at least once after treatment completion. For the spouses or partners, the inclusion criteria were: (1) living with Thai women with cervical cancer after complete treatment participants, and (2) being able to read and write in the Thai language.

The sample size was determined using the G*Power software version 3.1.9.4. The researcher calculated the power based on the number of participants recruited for the study and found an effect size of 0.26 [25] with 7 predictors and a sample size of 100. The calculated power was 0.97, indicating that a sample size of 100 couples was adequate for this study [26].

### 2.3. Research Instrument and Validation

#### 2.3.1. The Demographic Questionnaire

The researcher developed the demographic questionnaire. The demographic data for both Thai cervical cancer after treatment at least one year (TCCATALOY) and their spouses included age, religion, education, occupation, duration of marriage, number of children, and any underlying diseases. Information about disease and treatment of TCCATALOY were collected from medical records, which included the stage of the disease, types of treatment received, and any comorbidity/underlying diseases.

#### 2.3.2. The Thai Hospital Anxiety and Depression Scale (Thai HADS)

The Hospital Anxiety and Depression Scale (HADS), developed by Zigmond and Snaith in 1983 and translated into the Thai language in 1996 [27,28], consisted of 14 items—7 for anxiety and 7 for depression—rated on a 0–3 Likert scale, with a maximum score of 21 for each. A pilot test with 10 couples yielded a Cronbach’s alpha of 0.77 for anxiety and 0.73 for depression in women with cervical cancer, and 0.90 for anxiety and 0.82 for depression among their spouses.

#### 2.3.3. The Charlson Comorbidity Index (CCI)

The Charlson Comorbidity Index (CCI) was developed by Charlson and colleagues in 1987 [29]. It has been translated into Thai and shown to have accepted validity [30]. The CCI scores for disease range from 0 to 33, with interpretations as follows: “0 = no comorbidities”, “1–2 = few comorbidities”, “3–4 = moderate comorbidities”, and “>4 = a lot of comorbidities”.

#### 2.3.4. Couple Relationship Scale: CRS

The Couple Relationship Scale (CRS) was developed by Anderson, Johnson, Miller, and Barham in 2021 [31]. The CRS is a 10-item test of relational functioning that evaluates physical intimacy, conflict, coherence, safety, emotional intimacy, commitment, trust, and overall happiness and personal well-being [31]. Each dimension is scored from 0 to 100, and the total score is divided by the number of dimensions, with a cutoff score of 70.9 indicating clinical levels of relationship distress, with lower scores corresponding to greater levels of distress in the relationship [31]. In this study, after obtaining permission, the questions were translated into Thai using a back-translation method by the author. The Thai version was then translated back into English by a different translator and returned to the original developer to ensure the comprehensiveness, clarity, appropriateness, and cultural relevance of the instrument [32]. The item-level content validity index (CVI) from five experts for the instruments was 0.93. Reliability was assessed through a pilot test involving 10 couples of women with cervical cancer after treatment and their spouses, using Cronbach’s alpha coefficient. The Cronbach’s alpha coefficient was 0.98 for women with cervical cancer and 0.95 for spouses.

#### 2.3.5. Natsal 2000

Natsal 2000 [33] was developed by the National Survey of Sexual Attitudes and Lifestyles in 2000 [33]. It consists of 7 items (1–5 items for both women and men, item 6 for men, and item 7 for women), with ‘yes’ or ‘no’ answers. High scores indicate high sexual difficulties. In this study, the questions were translated into Thai using a back–translation method. The CVI from five experts for the instruments was 1. Reliability was assessed through a pilot test of 10 couples. The Cronbach’s alpha coefficient of the Natsal 2000 questionnaire yielded Kuder–Richardson Formula 20 (KR-20) scores of 0.47 for women with cervical cancer and 0.56 for their spouses. It is noticed that the coefficient of this questionnaire is quite low even though it passed the feasibility study in 900 respondents by the developer in 1997, which may be attributed to cultural differences between Western and Eastern countries [33].

#### 2.3.6. The New Sexual Satisfaction Scale (NSSS)

The New Sexual Satisfaction Scale (NSSS), developed by Stulhofer and colleagues in 2010, measures sexual satisfaction through three dimensions: pleasure (sexual sensation and sexual presence/awareness), satisfaction (sexual exchange and emotional connection/closeness), and orgasm (sexual activity) [34]. It consists of two subscales: partner/sexual activity-centered and the ego-centered subscale, comprising a total of 20 items. Respondents rate their satisfaction with their sex life over the past six months on a 5-point Likert scale, where higher scores indicate high levels of sexual pleasure. The Ego-Centered subscale includes items 1–10, while items 11–20 belong to the Partner and Activity-Centered subscale. In this study, the NSSS was translated into Thai using a back-translation method, achieving a CVI of 0.87 from five experts. Reliability was assessed through a pilot test involving 10 couples, resulting in Cronbach’s alpha coefficient of 0.98 for women with cervical cancer and 0.93 for their spouses. Questions were used and translated into the Thai language using the back-translation technique.

## 3. Results

Out of the 684 patients screened, 581 (84.94%) were excluded (see Figure 2). The most common reasons for exclusion were the lack of a spouse (52%) and no SI (32%). Of the remaining 103 couples, 3 refused to participate due to a lack of interest from either the patients or the spouses. Finally, 100 patients and spouses provided baseline demographic information and agreed to participate in the study.

Demographic data showed that the mean age of women with cervical cancer who had completed treatment at least one year of treatment was 47.79 (SD = 8.57). The mean duration of marriage was 19.40 (SD = 11.83, ranging from 2 to 46 years). Table 1 presented that over half were diagnosed with stage II cancer (52%) and 48% received concurrent chemoradiotherapy (CCRT). The mean age of their spouses was 50.62 (SD = 9.74). Please see Table 1 for details.

The average total score for sexual difficulties was 2.59 for women treated for cervical cancer for at least one year, and 1.02 for their spouses (SD = 1.71 and 1.41, respectively), with scores ranging from 0 to 6 for the women and 0 to 5 for the spouses. Specific items related to sexual difficulties are presented in Table 2. Additionally, women reported lubrication issues as their most prominent difficulty, while their spouses expressed anxiety about performance as the greatest concern. Notably, lubrication problems were common, as CCRT can lead to vaginal dryness [19,35,36,37,38]. Additionally, during treatment, patients are often advised to avoid sexual activity, which can cause anxiety for their spouses, who may be unsure how to engage in sex after the treatment. Please see Table 2 for details.

The average couple relationship score for women treated for cervical cancer for at least one year and their spouses were 79.64 and 83.36, respectively (SD = 19.04 and 13.77), ranging from 10 to 100 and 17 to 100, respectively. The specific items related to the couple’s relationship are shown in Table 3. Notably, women scored highest in acceptance, while spouses scored highest in safety. Please see Table 2 for details.

Pearson correlation analysis showed that couple relationships were positively correlated with sexual satisfaction, while age, anxiety, depression, comorbidities, marriage duration, and sexual difficulties were negatively correlated for both women and their spouses. In women, significant correlations included anxiety, depression, marriage duration, couple relationships, and sexual difficulties, with sexual difficulties showing a moderate correlation. In spouses, anxiety, depression, comorbidities, couple relationships, and sexual difficulties were significant, with sexual difficulties showing a strong correlation, and anxiety, depression, and couple relationships showing moderate correlations. Please see Table 3 for details.

Multiple linear regression analysis revealed that sexual difficulties (b = −2.82, *p* = 0.005) and marriage duration (b = −0.48, *p* = 0.012) significantly predicted sexual satisfaction in women treated for cervical cancer (Table 4), while sexual difficulties (b = −3.13, *p* = 0.004) and couple relationships (b = 0.30, *p* = 0.003) were significant predictors for their spouses. These factors accounted for 25.6% of the variance in women’s sexual satisfaction and 34.9% in their spouses. For details, see Table 5.

These results suggest that reduced sexual satisfaction can be challenges for couples, as it may negatively affect both intimacy and sexual function. The interaction of the three predictors highlights the appropriate interventions to better improve sexual satisfaction for both patients and their spouses.

## 4. Discussion

This study found that sexual difficulties and the duration of marriage accounted for 25.6% of patients’ sexual satisfaction, with marriage duration being the strongest predictor to sexual satisfaction. In contrast, sexual difficulties and the couple’s relationship predicted 34.9% of spouses’ sexual satisfaction, with sexual difficulties being the strongest predictor.

Regarding sexual difficulties, our findings align with previous research on gynecological cancer survivors in the United States, which shows that these individuals often lose interest in sex due to feelings of insecurity, causing them to hesitate or avoid sexual activities [8]. A similar finding has been observed among cervical cancer survivors in Australia, where pain during sexual intercourse is strongly linked to lower sexual satisfaction [20]. In our study, half of the women had stage II cervical cancer and received CCRT without surgery, and prior studies show that the radiation could affect the remaining cancerous tissue as well as the vaginal area and its lining, causing pain during sexual intercourse [19,35,36,37,38]. Notably, early-stage (stage I) cervical cancer typically presents no symptoms, as it has formed in the cervix only; as a result, women may not seek medical check-ups [39,40,41]. However, in stage II, when the cancer spreads to the upper vagina or surrounding tissue, leading to symptoms like abnormal bleeding (i.e., after sex, post menopause, or between periods) typically occur [39,40,41]. This prompts women to seek medical attention as reflected in our study, where half of the women were diagnosed at stage II. This underscores the significance of regular screening in enhancing survival rates, improving outcomes and managing treatment side effects effectively [37]. Among the side effects of treating cervical cancer, sexual difficulties are commonly observed [37]. Our study found that about 57% of patients reported sexual difficulties related to lubrication, while 17% of their spouses voiced concerns about their sexual performance. Our findings are consistent with previous studies in Ghana, India, the Netherlands, and the United States, where women with cervical cancer experienced difficulties with lubrication and pain during intercourse [37,42]. The high rate of lubrication related to sexual difficulties in our study aligns with global reports, which show a range of 17–58% [37,43]. This underscores the universal challenges faced by couples after cervical cancer treatment, highlighting the need for both patient and spouse preparation, along with a multidisciplinary approach to support them. On the other hand, their partners reported issues like premature ejaculation, trouble getting or keeping an erection, and a reduced interest in sex [37,42]. Therefore, this highlights the importance of training healthcare personnel to support both patients and their spouses in managing issues that may arise after cervical cancer treatment.

Our finding also found that the duration of marriage was the strongest predictor of sexual satisfaction among patients with cervical cancer. This finding is consistent with earlier studies indicating that sexual satisfaction tends to decline as the length of marriage increases [24,44]. A similar study was found in Spain among healthy men and women, as well as with young adults in Poland and Germany [24,44,45]. In addition, a previous study conducted in 19 countries in Asia, Africa, and the Americas found that longer marriage is associated with lower levels of sexual satisfaction [21]. According to previous studies, sexual desire and satisfaction are often elevated in the early stages of a relationship. Nonetheless, as the relationship becomes more established and more long-term, sexual satisfaction may decline, as sexual activities tend to become more routine and habitual over time [21,24,46,47,48]. In our study, the average duration of marriage was 19 years; however, we did not have data on whether sexual activity had already declined before the diagnosis of cervical cancer or if it happened afterward. Thus, this highlights the need for collective and more comprehensive information to better assess and address the patient’s needs and concerns.

In contrast, the couple’s relationship was found to be the strongest predictor of the spouses’ sexual satisfaction. This finding aligns with an earlier study involving older men in Britain, which indicated that relationship satisfaction was linked to their sexual satisfaction [49]. A similar result was found among new couples in the Midwest of the United States, where higher relationship satisfaction was linked to higher sexual satisfaction [50]. A previous study showed that both women and men who perceive their relationship as positive are more likely to be satisfied with their sexual lives [51]. Indeed, the feeling of sexual desire is more likely to be intended during the earlier of dating, and their interest in sexual desire could decline after a few years of dating, which ultimately links to decreased sexual satisfaction [52]. However, a prior study indicated that sexual desire in couples can improve when they perceive their relationship positively, develop trust, and enhance intimacy [42]. Based on ecological theory, sexual satisfaction is considered part of the mesosystem and can be influenced by the overall relationship context [17]. Therefore, healthcare providers should pay attention to spouses of individuals with cervical cancer. These spouses need to be informed about the treatment process, its potential side effects, and how it might impact their relationship as a couple.

In terms of demographics and clinical data, we found that factors such as age, comorbidities, anxiety, and depression did not have a significant impact on sexual satisfaction for both patients and their spouses. Similar results were reported in Germany and the United States, where age did not impact sexual satisfaction in adult populations and older married women [53,54]. In contrast, studies from Portugal, Slovenia, Croatia, Bosnia, and Romania indicated that age was a significant factor, with sexual satisfaction generally declining as people get older [22,55]. Similarly, a study in Egypt showed a significant decline in sexual satisfaction among married women over the age of 50 [23]. In comparison, our study involved patients who were slightly younger, with an average age of 47; however, these patients had undergone cervical cancer treatment, which can lead to early menopause. This, in turn, can cause a decrease in hormone levels, reduced sexual interest, increased sexual difficulties, and overall sexual dissatisfaction. Regarding comorbidities, our findings differ from studies in Germany and China, where a higher number of comorbidities, such as heart disease and musculoskeletal problems, were significantly linked to lower sexual satisfaction [19,53]. However, in our study, the majority of the patients and their spouses reported having few or no comorbidities; thus, making it difficult to determine whether comorbidities had any impact on sexual satisfaction. In terms of anxiety and depression, our results differ from earlier studies that found a connection to sexual satisfaction [19,51], given we measured anxiety and depression in both patients and spouses using a 14-item questionnaire, which asked general questions about their experience with anxiety and depression over the past week. However, this approach may not have fully captured their actual levels of anxiety and depression, especially related to treatment side effects. Additionally, sexual problems may not have been explored in depth, as no specific questions were included. According to our findings, most patients (77%) and spouses (90%) reported no anxiety, and 93% of couple reported no depression. As a result, we could not determine if anxiety and depression actually affect sexual satisfaction. Future studies should use more specific measurement tools, with questions linked to sexual satisfaction, and consider the best timing for exploring these issues.

Our findings, which show that sexual difficulties and marriage duration are key predictors of sexual satisfaction for women, while sexual difficulties and the quality of the couple’s relationship are key for spouses, align with the ecological theory. This theory demonstrates how components within the mesosystem interact and influence each other, effecting sexual satisfaction trajectories. Regarding the microsystem, although factors such as age, depression and anxiety, and physical function (in our study, comorbidity) were not significant predictors of sexual satisfaction, these mixed results are consistent with the ecological framework, where similar patterns have been observed in other studies [18,21].

## 5. Conclusions

This study explored factors influencing sexual satisfaction among Thai women with cervical cancer one year after treatment, as well as their spouses’ perspectives. The findings are important for nursing practice, as they highlight the important of early counseling through constructive communication for both patients and spouses during follow-up care. Future research should focus on developing culturally tailored interventions to reduce sexual stigma and promote the use of telehealth to improve access, encourage couples to address sexual issues, and offer them ongoing support.

This study has some limitations. First, the cross-sectional design does not allow us to establish causal relationships, so longitudinal and qualitative studies are needed to examine changes over time and gain deeper insights into couples’ experiences post cervical cancer treatment. Second, the use of self-report questionnaires may lead to inaccurate recall or social desirability bias; therefore, objective measures (i.e., clinical assessment, health record) should be incorporated to minimize these potential issues. Third, the small sample size indicates the results should be interpreted with caution as they limit the generalizability of the results to population beyond urban Thailand and may not apply to a broader group, so future studies should include a larger sample size to improve statistical power. Lastly, the exclusion of women without spouses and those who did not engage in sexual intercourse following treatment restricts the generalizability to these groups, so the results should be interpreted carefully.

## Figures and Tables

**Figure 1 healthcare-13-00112-f001:**
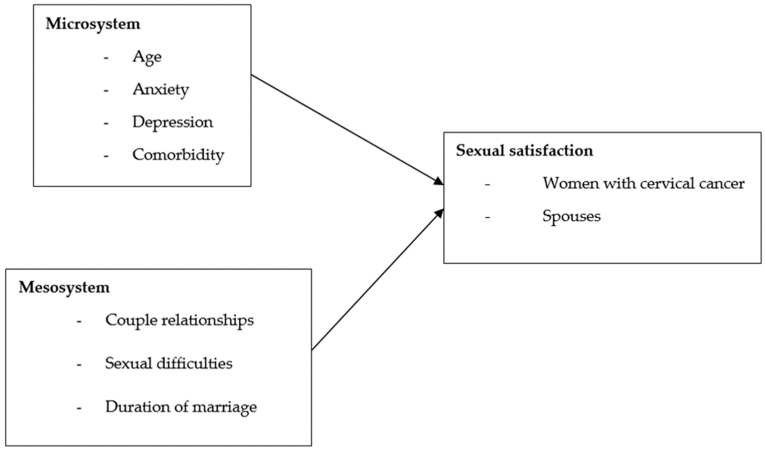
Conceptual framework of the study derived from ecological theory.

**Figure 2 healthcare-13-00112-f002:**
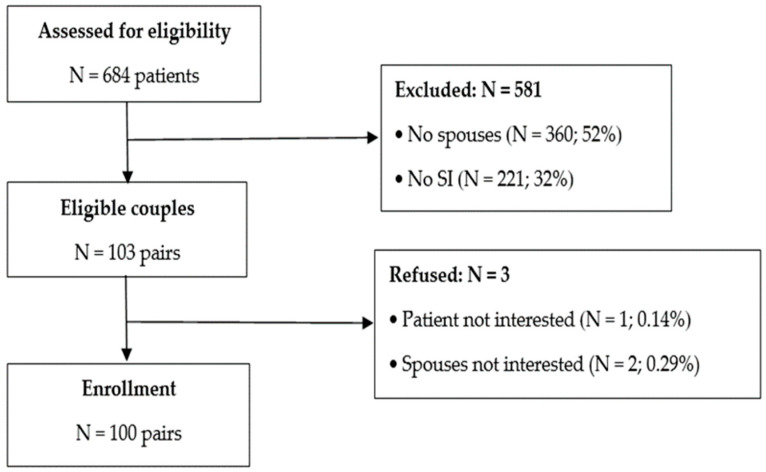
Study consort diagram.

**Table 1 healthcare-13-00112-t001:** Demographics and clinical characteristics.

Characteristics	Patients N (%) or Mean ± SD	Spouses N (%) or Mean ± SD
Microsystem Level		
**Age** (in years)	47.79 ± 8.57	50.62 ± 9.74
**Stage of disease**		
I	35 (35)	
II	52 (52)	
III	13 (13)	
**Treatments**		
Surgery	32 (32)	
Concurrent chemoradiotherapy	48 (48)	
Surgery + Concurrent chemoradiotherapy	15 (15)	
Surgery + Radiotherapy	5 (5)	
**Anxiety**		
Normal	77 (77)	90 (90)
Borderline case	18 (18)	5 (5)
Case	5 (5)	5 (5)
**Depression**		
Normal	93 (93)	93 (93)
Borderline case	6 (6)	6 (6)
Case	1 (1)	1 (1)
**Comorbidities** (in scores)	2.41 ± 1.11	0.22 ± 0.52
**Mesosystem level**		
**Duration of marriage** (in years)	19.40 ± 11.83	
**CRS**		
Good relationship	78 (78)	86 (86)
Bad relationship	22 (22)	14 (14)

**Table 2 healthcare-13-00112-t002:** The items of sexual difficulties and CRS reported by patients and their spouses.

Sexual Difficulties *			
Items	Statements	Patients (%)	Spouse (%)
1	Lack of interest in sex	34	14
2	Anxious about performance	52	17
3	Unable to experience orgasm	33	10
4	Premature orgasm	20	15
5	Painful intercourse	37	2
6	Unable to achieve or maintain erection (men only)		14
7	Trouble lubricating (women only)	57	
**CRS** **			
**Items**	**Dimensions of the CRS**	**Patients** (M, SD)	**Spouse** (M, SD)
1	Emotional Intimacy	(76.22, 26.19)	(81.78, 18.45)
2	Commitment	(78.09, 23.56)	(82.45, 18.45)
3	Trust	(82.61, 23.52)	(85.26, 15.19)
4	Safety	(82.41, 24.57)	(85.79, 16.22)
5	Cohesion	(79.79, 24.73)	(83.85, 17.96)
6	Acceptance	(84.21, 19.80)	(85.07, 15.86)
7	Conflict	(80.29, 22.56)	(83.53, 16.75)
8	Physical Intimacy	(76.68, 24.62)	(81.53, 19.41)
9	Overall Happiness in Relationship	(78.95, 21.78)	(82.47, 19.96)
10	General Personal Well-being	(77.13, 21.65)	(81.89, 17.64)

* Sexual difficulties: item 6, men only; item 7, women only. ** CRS: Couple Relationship Scale.

**Table 3 healthcare-13-00112-t003:** Pearson correlation analysis of factors predicting the sexual satisfaction of patients (N = 100) and their spouses (N = 100).

Pearson Correlation	Sexual Satisfaction	Age	Sexual Difficulties	Duration of Marriage	Comorbidities	Couple Relationship	Depression	Anxiety
Sexual satisfaction (W)	1.00	−0.10	−0.41 *	−0.22 *	−0.02	0.26 *	−0.20 *	−0.22 *
Sexual satisfaction (S)	1.00	−0.13	−0.50 *	−0.16	−0.27 *	0.39 *	−0.32 *	−0.35 *
Age (W)		1.00	−0.05	0.73	0.27	0.20	−0.10	−0.22
Age (S)		1.00	0.06	0.73	0.31	0.25	0.06	0.03
Sexual difficulties (W)			1.00	0.00	−0.02	−0.42	0.29	0.33
Sexual difficulties (S)			1.00	0.03	0.36	−0.38	0.51	0.57
Duration of marriage (W)				1.00	0.28	0.19	−0.14	−0.24
Duration of marriage (S)				1.00	0.26	0.26	−0.07	−0.05
Comorbidities (W)					1.00	0.02	0.13	0.07
Comorbidities (S)					1.00	−0.11	0.40	0.39
Couple relationship (W)						1.00	−0.43	−0.37
Couple relationship (S)						1.00	−0.32	−0.25
Depress (W)							1.00	0.64
Depress (S)							1.00	0.83
Anxiety (W)								1.00
Anxiety (S)								1.00

* *p*-value < 0.05.

**Table 4 healthcare-13-00112-t004:** Multiple linear regression analysis of sexual satisfaction among patients with cervical cancer (N = 100).

Variables	b	SE	Beta	Adjusted R^2^	*t*	*p*-Value
Age	0.12	0.25	0.06	0.00	0.48	0.63
Anxiety	−0.72	0.64	−0.14	0.04	−1.12	0.27
Depression	−0.13	0.73	−0.02	0.03	1.18	0.86
Comorbidities	1.03	1.41	0.07	−0.01	0.73	0.47
Sexual difficulties	−2.82	0.97	−0.30	0.16	−2.89	0.005
Couple relationship	0.11	0.09	0.13	0.06	1.17	0.25
Duration of marriage	−0.48	0.19	0.35	0.04	−2.58	0.012

Constant = 67.37, R^2^ = 0.256, SE = Standard Error, b = unstandardized coefficient, Beta = regression coefficient.

**Table 5 healthcare-13-00112-t005:** Multiple linear regression analysis of sexual satisfaction among spouses (N = 100).

Variables	b	SE	Beta	Adjusted R^2^	*t*	*p*-Value
Age	−0.03	0.18	0.02	0.01	−0.18	0.86
Anxiety	−0.43	0.62	−0.11	0.11	−0.70	0.49
Depression	0.18	0.80	0.04	0.10	0.23	0.82
Comorbidities	−0.66	2.59	−0.03	0.07	−0.25	0.80
Sexual difficulties	−3.13	1.06	−0.33	0.25	−2.95	0.004
Couple relationship	0.30	0.10	0.30	0.14	3.04	0.003
Duration of marriage	−0.23	0.15	−0.20	0.02	−1.59	0.12

Constant = 54.29, R^2^ = 0.349, SE = Standard Error, b = unstandardized coefficient, Beta = regression coefficient.

## Data Availability

The data presented in this study are available on request. The data are not publicly available due to patient privacy.

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
