# Peer review of "Factors Predicting Sexual Satisfaction of Thai Women with Cervical Cancer After Complete Treatment at Least One Year and Their Spouses"

_healthcare, 2025, doi:10.3390/healthcare13020112_

Round 1

Reviewer 1 Report

Comments and Suggestions for Authors

Title and Abstract

  • The title is appropriate, but the abstract lacks details about the theoretical framework and practical implications.
  • Action
    1. Explicitly mention the application of ecological theory in the abstract.
    2. Include a practical implication, such as how the findings can inform "couple-based communication enhancement interventions."

Introduction

    1. The introduction should be updated with global and regional cervical cancer statistics, particularly for Southeast Asia.
    2. The rationale for using ecological theory is not clear, and the cultural nuances of sexual satisfaction in Thailand are underexplored.
  • Action
    1. Add regional cervical cancer statistics and survival rates, e.g., "Cervical cancer contributes to approximately X% of gynecological cancers in Southeast Asia."
    2. Clarify the selection of ecological theory and focus on how the microsystem and mesosystem are suitable for analyzing relational dynamics.
    3. Briefly discuss cultural norms in Thailand, such as attitudes toward sexuality and intimacy in marital relationships.
    4. Expand on ecological theory, detailing why the microsystem and mesosystem levels are critical for understanding sexual satisfaction e.g Explain how cultural norms in Thailand influence couple dynamics and sexual well-being

Theoretical Framework

    1. The manuscript focuses on ecological theory but does not incorporate the bio-psycho-social model or adequately emphasize the dyadic nature of sexual satisfaction.
  • Action
    1. Highlight how the bio-psycho-social model interacts with ecological theory, e.g.,
    2. Justify the inclusion of spousal perspectives, e.g., Couples' satisfaction is interdependent and this necessitate a dyadic approach.

Methods

    1. The justification for sample size is insufficient.
    2. Cultural considerations and mitigation of self-report biases are not addressed.
  • Action
    1. Provide a stronger rationale for the chosen sample size and reference statistical power analysis.
    2. Explain how cultural norms and potential self-report biases were mitigated, e.g., any training on cultural sensitivity?
    3.  Elaborate on the validation of translated questionnaires, particularly the reliability of the Natsal 2000, which shows low internal consistency. Example: Discuss pilot testing outcomes and improvements made to the instrument. Action: Include details on how ethical concerns, particularly around participant privacy and sensitivity in discussing sexual satisfaction, were addressed

Results

Figures and Tables

    1. Figures and tables are detailed but can overwhelm readers with excessive information.
  • Action:
    1. Simplify Figure 1 by focusing on the microsystem and mesosystem.
    2. Combine Tables 2 and 3 to streamline information about sexual difficulties and couple relationships.

Discussion

    1. Results are detailed but does not have a clear connection to ecological theory.
    2. Demographic trends such as the high proportion of stage II cervical cancer are not sufficiently explained.
    3. The discussion provides limited actionable implications and cultural context.
    4. It does not compare findings with global studies in enough depth.

Action

    1. Relate findings to ecological theory, e.g., These results align with the mesosystem level and emphasize relational dynamics.
    2. Briefly discuss surprising trends, e.g., The prevalence of stage II cancer reflects a potential delay in diagnosis due to limited screening access.
    3. Expand on culturally tailored interventions, e.g., "Thai-specific counselling strategies could address sexual stigmas."
    4. Compare with global findings, e.g., "Our findings on lubrication issues align with studies in Ghana and Australia and reflect universal post-treatment challenges."

Limitations

    1. Limitations lack specificity regarding generalizability and cross-sectional design constraints.
  • Action: Add specifics on the small sample size which limits generalizability of findings beyond urban Thai populations.
    1. Emphasize the need for longitudinal studies to examine changes over time.

Conclusion

    1. The conclusion does not sufficiently address future research directions.
  • Suggest concrete areas for future studies, e.g., "Future research should explore longitudinal trends in sexual satisfaction and test telehealth interventions for couples."

Comments on the Quality of English Language

The English could be improved to more clearly express the research.

Author Response

Dear reviewer,

I attached file for revision to you

Best regard,

Wunwisa Bualoy

Reviewer 2 Report

Comments and Suggestions for Authors

I would like to commend the authors for an excellent and well-executed study. The manuscript is well-structured, insightful, and addresses an important yet underexplored topic—sexual satisfaction among Thai women with cervical cancer after treatment and their spouses. The use of ecological theory provides a strong theoretical foundation, and the statistical analysis is robust and appropriate for the study's objectives.

The findings are clearly presented and offer valuable implications for healthcare providers, particularly in supporting both patients and their spouses post-treatment. The study contributes meaningful knowledge to the field of post-cancer care and relationship dynamics.

In its current form, the paper is clear, well-written, and of high quality. I have no significant concerns and recommend the manuscript for publication.

Author Response

(The authors gave the same response as above.)

Reviewer 3 Report

Comments and Suggestions for Authors

Dear Authors,

Thank you for the opportunity to review your manuscript, “Factors Influencing Sexual Satisfaction in Thai Women with Cervical Cancer and Their Partners.” This study addresses a valuable and underexplored area, offering insights into the complex interplay of sexual satisfaction, relationship factors, and cancer survivorship. Your work has the potential to guide clinical practice, informing interventions that support both patients and their partners. The methodological rigor in translation and the use of validated instruments (e.g., Thai HADS, CRS, NSSS) is commendable, and the study’s ecological framework helps situate sexual satisfaction within a broader relational and cultural context.

There are, however, areas that would benefit from further refinement. More explicit justification of the cross-sectional design, clearer discussion of the low statistical power, deeper analysis of the implications of low instrument reliability, and a stronger linkage between findings, theory, and clinical practice would enhance the manuscript’s clarity and impact.

Title and Abstract

The title accurately captures the core focus of the study. The abstract effectively summarizes the objectives, methodology, and principal findings. To strengthen this section, please include explicit mention of the primary predictors of sexual satisfaction identified for both women (sexual difficulties and duration of marriage) and partners (sexual difficulties and couple relationship). This will ensure that readers grasp the key outcomes and their implications from the outset.

Introduction

The introduction provides a solid overview of the importance of understanding sexual satisfaction in the context of cervical cancer survivorship. Your reference to ecological theory is valuable, but consider articulating more explicitly how this theoretical lens guides your hypotheses and interpretation of results. For instance, clarify how individual, relational, and contextual factors interact to shape sexual satisfaction. Additionally, clearly state your study objectives and hypotheses. This will help readers anticipate your methodological choices and understand the significance of your findings in relation to existing literature.

Materials and Methods

You have clearly described the descriptive, cross-sectional study design. To strengthen this section, briefly explain why a cross-sectional approach is suitable for these research questions and acknowledge that this design precludes causal inferences. Consider adding a sentence on how future longitudinal or mixed-method studies could build on these findings.

Your inclusion and exclusion criteria are transparent, though excluding women without partners or those without recent sexual activity may limit the generalizability of the results. It would be helpful to note this limitation explicitly and discuss how it might influence the applicability of your conclusions, especially in broader or more diverse populations.

Please address the reported power of .097. While the sample size may be practically constrained, explaining why this power level was deemed acceptable, or discussing its implications for interpreting effect sizes, would provide readers with greater confidence in your analyses.

The use of validated instruments is commendable. However, consider providing more detail on the low reliability (KR-20 of .47 and .56) of the Natsal 2000 scale. Discuss possible reasons (e.g., cultural adaptation challenges, sensitivity of sexual topics) and how this may have influenced the measurement of sexual difficulties. If alternative measures or additional validation steps are planned for future research, note this here.

Results

Your results are clearly organized and presented with well-structured tables and figures. Ensure each figure and table is referenced in the text for seamless reading. The correlation and regression analyses are appropriate; however, offering a brief explanation of the practical significance of your statistically significant predictors—beyond reporting p-values and coefficients—would help readers gauge the clinical or relational importance of these findings. For instance, highlight what a specific decrease in sexual satisfaction might mean in a practical, day-to-day context for couples managing life after cancer treatment.

Discussion

The discussion effectively situates the findings within existing literature. You identified that sexual difficulties and marriage duration are key predictors for women, while sexual difficulties and the quality of the couple’s relationship are key for partners. This insight is valuable—consider linking it more explicitly to the ecological framework mentioned in the introduction. For example, explain how relational and individual factors interact within the ecological model to shape sexual satisfaction trajectories.

When discussing clinical implications, provide concrete recommendations for healthcare providers. For instance, suggest integrating routine assessments of sexual health into survivorship care, offering tailored counseling sessions focused on communication and intimacy, or designing interventions to manage pain, lubrication issues, and performance anxiety. Such actionable steps will help readers understand how to translate these findings into practice.

Your finding that most participants reported low levels of anxiety and depression is intriguing. Reflect on whether this could be due to cultural factors, measurement timing, or sample characteristics, and consider how future research might capture a more nuanced psychological profile.

Limitations

The limitations are acknowledged, including the cross-sectional design, reliance on self-report measures, and the relatively small sample size. To strengthen this section, suggest how future studies might address these constraints. For example, recommend longitudinal designs to track changes in sexual satisfaction over time, larger samples to improve statistical power and generalizability, and possibly qualitative methods to gain deeper insight into couples’ lived experiences.

Conclusions

Your conclusions summarize the key findings well. Reiterate how these findings relate back to the study’s original objectives and hypotheses, explicitly highlighting how the ecological perspective aids in understanding the multifaceted nature of sexual satisfaction. Emphasize the importance of early counseling, tailored interventions, and consideration of relational context. This will reinforce the value of your contributions to both theory and practice.

I look forward to seeing your revised manuscript.

Best regards,

Author Response

(The authors gave the same response as above.)

Round 2

Reviewer 1 Report

Comments and Suggestions for Authors

The authors have addressed nearly all major concerns by clarifying the theoretical framework, expanding on ecological theory, and incorporating culturally specific insights. The methods section is more transparent, and the discussion now better connects findings to practice and existing literature. While the sample size remains a limitation, the revisions substantially strengthen the paper’s clarity, rigor, and applicability.

Reviewer 3 Report

Comments and Suggestions for Authors

The article has improved significantly, and I am satisfied with its current version. The revisions have effectively addressed the key points, and the manuscript is now well-structured and provides valuable insights. I believe it makes a meaningful contribution to the field.